# Potential Prognostic Value of GATA4 Depends on the p53 Expression in Primary Glioblastoma Patients

**DOI:** 10.3390/genes14061146

**Published:** 2023-05-25

**Authors:** Berenika Trąbska-Kluch, Marcin Braun, Magdalena Orzechowska, Sylwia Paszek, Alina Zuchowska, Julia Sołek, Adam Kluska, Jacek Fijuth, Dorota Jesionek-Kupnicka, Izabela Zawlik

**Affiliations:** 1Department of Teleradiotherapy, Copernicus Memorial Hospital, 93-513 Lodz, Poland; berenika.trabska@gmail.com (B.T.-K.); jacek.fijuth@umed.lodz.pl (J.F.); 2Department of Radiotherapy, Chair of Oncology, Medical University of Lodz, 93-513 Lodz, Poland; 3Department of Pathology, Chair of Oncology, Medical University of Lodz, 93-513 Lodz, Poland; braunmarcin@gmail.com (M.B.); juliasolek@gmail.com (J.S.); dorota.jesionek-kupnicka@umed.lodz.pl (D.J.-K.); 4Department of Molecular Carcinogenesis, Medical University of Lodz, 93-513 Lodz, Poland; magdalena.orzechowska@umed.lodz.pl; 5Laboratory of Molecular Biology, Centre for Innovative Research in Medical and Natural Sciences, University of Rzeszow, 35-959 Rzeszow, Poland; sylwia.paszek@wp.pl (S.P.); alzuch4@gmail.com (A.Z.); 6Department of General Genetics, Institute of Medical Sciences, College of Medical Sciences, University of Rzeszow, 35-959 Rzeszow, Poland; 7Brachytherapy Department, Greater Poland Cancer Centre, 61-866 Poznan, Poland; adamkluska88@gmail.com

**Keywords:** glioblastoma, *GATA4* promoter methylation, GATA4 protein expression, TP53 abnormalities

## Abstract

Background: Primary glioblastoma is characterized by an extremely poor prognosis. The promoter methylation of *GATA4* leads to the loss of its expression in many cancer types. The formation of high-grade astrocytomas can be promoted by the concurrent loss of *TP53* and GATA4 in normal human astrocytes. Nevertheless, the impact of *GATA4* alterations with linkage to *TP53* changes in gliomagenesis is poorly understood. This study aimed to evaluate GATA4 protein expression, *GATA4* promoter methylation, p53 expression, *TP53* promoter methylation, and mutation status in patients with primary glioblastoma and to assess the possible prognostic impact of these alterations on overall survival. Materials and Methods: Thirty-one patients with primary glioblastoma were included. GATA4 and p53 expressions were determined immunohistochemically, and *GATA4* and *TP53* promoter methylations were analyzed via methylation-specific PCR. *TP53* mutations were investigated via Sanger sequencing. Results: The prognostic value of GATA4 depends on p53 expression. Patients without GATA4 protein expression were more frequently negative for *TP53* mutations and had better prognoses than the GATA4 positive patients. In patients positive for GATA4 protein expression, p53 expression was associated with the worst outcome. However, in patients positive for p53 expression, the loss of GATA4 protein expression seemed to be associated with improved prognosis. *GATA4* promoter methylation was not associated with a lack of GATA4 protein expression. Conclusions: Our data indicate that there is a possibility that GATA4 could function as a prognostic factor in glioblastoma patients, but in connection with p53 expression. A lack of GATA4 expression is not dependent on *GATA4* promoter methylation. GATA4 alone has no influence on survival time in glioblastoma patients.

## 1. Introduction

Glioblastoma multiforme (GBM, WHO, IV) is the most frequent primary brain tumor in adults and is characterized by very rapid growth and an extremely short overall survival time. It is classified as either a primary—de novo—or secondary tumor that develops as a result of the progression of lower-grade malignant gliomas. The standard treatment involves surgical resection followed by radiochemotherapy (RT-CHT) and adjuvant chemotherapy (CHT) with temozolomide (TMZ). However, GBM remains an incurable disease, with a median survival time of 15 months in patients who underwent multimodality treatment [1]. The unfavorable prognosis in GBM is caused by the high heterogeneity of this tumor group and its resistance to therapy [2]. Primary glioblastoma differs from secondary GBM not only in its formation, but also in the characterization of molecular pathways [3].

Epigenetic alterations are frequent occurrences in glioblastoma pathogenesis. Epigenetics is a mechanism of gene expression regulation that is important for development and cell proliferation without any alterations in gene sequence. The identification of epigenetic modifications seems to be crucial in explaining glioma etiopathogenesis and can indicate their prognostic value. One of the epigenetic mechanisms is gene promoter methylation, which causes gene silencing. Among the genes silenced in GBM due to its promoter hypermethylation is *GATA4*, which encodes a transcription factor involved in the regulation of apoptosis [4]. It has been shown that the hypermethylation of *GATA4*’s promoter leads to the loss of its expression, which has been observed in several types of cancers, such as colorectal, gastric, and lung cancers [5,6,7]. Agnihotri et al. demonstrated that the loss of GATA4 protein expression was a negative survival prognostic factor in GBM patients and that GATA4 can sensitize GBM cells to alkylating drugs [4]. It has also been revealed that the loss of GATA4 in normal human astrocytes leads to high-grade astrocytoma formation in conjunction with other important genetic modifications, such as the loss of *TP53* [4]. *TP53* has been found to demonstrate abnormalities in GBM, and its regulation plays a significant role in gliomagenesis. It also has an important role in the maintenance of genomic stability, the regulation of the cell cycle, cell differentiation, death, and responding to DNA damage [8]. *TP53* mutations have been detected in 28% to 58% of glioblastoma patients [9,10]. Compared to *IDH*-wildtype GBM, *IDH*-mutant GBM has a higher incidence of occurrence (27% vs. 81%) [11]. In GBM, *TP53* mutations are mostly point mutations, which not only abolish the tumor suppressor function, but confer gain-of-function (GOF) properties that promote GBM malignancy [12]. While *TP53* mutations can contribute to the development of GBM, additional genetic alterations, including mutations in *PTEN*, are necessary for the progression of this disease [13].

The relationship between GATA4 expression and *GATA4* promoter methylation and alterations in *TP53* and its effect on GBM development has not been defined well. The aim of this study was to evaluate GATA4 protein expression, *GATA4* promoter methylation, and TP53 status (mutations, promoter methylation, and protein expression) in patients with primary glioblastoma and assess the possible prognostic impact of these changes on the overall survival rate.

## 2. Materials and Methods

### 2.1. Patient Recruitment

Thirty-one patients (ten female and twenty-one male) with primary glioblastoma (GBM) diagnosed between 2006 and 2011 were recruited. All of their tumors were histopathologically examined and classified according to the WHO classification of tumors of the Central Nervous System (CNS) [14]. The formalin-fixed, paraffin-embedded samples were collected at the Department of Pathology, Chair of Oncology, Medical University of Lodz, Poland. The study was performed according to the Helsinki Declaration and the Institutional Review Board Committee. Ethics committee approval was obtained from the Institutional Review Board of the Medical University of Lodz (Number RNN/226/11/KE). All of the patients submitted informed, written consent to participate in the experiment.

### 2.2. DNA Isolation

Genomic DNA was isolated from formalin-fixed and paraffin-embedded tissue specimens of the patients using the QIAamp DNA FFPE Tissue Kit (Qiagen, Valencia, CA, USA), according to the manufacturer’s instructions. The quality and integrity of the obtained DNA was assessed via electrophoresis on 2% agarose gel. The DNA was stored at −20 °C.

#### 2.2.1. GATA4 Promoter Methylation Analysis via Methylation-Specific PCR

*GATA4* gene promoter methylation was analyzed using methylation-specific PCR (MSP). The isolated genomic DNA was modified with sodium bisulfide using the EpiTect Bisulfite Kit (Qiagen, Valencia, CA, USA), according to the manufacturer’s guidelines. Converted and purified DNA was stored at −80 °C before use. As a positive and negative control, EpiTect methylated/unmethylated human control DNA (bisulfite converted) (Qiagen, Valencia, CA, USA) were used. The reaction mixture (20 μL volume) contained 50–100 ng of bisulfite-treated DNA as a template, 0.5 U TaKaRaEpiTaq HS (TaKaRaClontech Laboratories, Inc., San Jose, CA, USA), 2.5 mM MgCl2, 0.3 mM dNTP Mixture, 1× EpiTaq PCR Buffer, and 0.3 µM of the primers. The amplification was performed in Labcycler 48 Gradient (SensoQuest Bio-medical electronics, Göttingen, Germany). The primer sequences were as follows: forward methylated-5′-GTATAGTTTCGTAGTTTGCGTTTAGC-3′; reverse methylated-5′-AACTCGCGACTCGAATCCCCG-3′ (product of 136 bp); forward unmethylated 5′- TTTGTATAGTTTTGTAGTTTGTGTTTAGT-3′; and reverse unmethylated 5′-CCCAACTCACAACTCAAATCCCCA-3′ (product of 142 bp).

The conditions for the GATA4-methylated amplicons were: 95 °C for 15 min, 38 cycles 95 °C for 30 s, 65 °C for 1 min, 72 °C for 1 min, and 72 °C for 7 min; in addition, for the GATA4-unmethylated amplicons, the conditions were: 95 °C for 15 min, 38 cycles 95 °C for 30 s, 62 °C for 1 min, 72 °C for 1 min, and 72 °C for 7 min. Each PCR reaction included a positive control (methylated DNA), negative control (unmethylated DNA), and water. Each PCR product (6 µL) was directly loaded onto 2% agarose gel and visualized under a UV illuminator (Syngen G: BOX). Methylation-specific PCR for *GATA4* was performed on 23 samples due to the unavailability of the material. Representative MSP results are shown in Figure 1.

#### 2.2.2. GATA4 Protein Expression Analysis via Immunohistochemistry

GATA4 protein expression was investigated immunohistochemically using a GATA4 monoclonal antibody (G-4 clone), 1:50 dilution, pH 8.0 (Santa Cruz Biotechnology, Santa Cruz, CA, USA), and processed in the EnVision system (DAKO, Glostrup, Denmark). GATA4 nuclear expression was defined as positive when the proportion of positive cells was greater than 10% (Figure 2A). Negative GATA4 immunostaining resulted in tumor sections (Figure 2B). The IHC results were verified using positive and negative tissue controls on all immunostaining reactions. Positive controls set up on human ovarian tissue showed nuclear staining of the follicle and ovarian stromal cells; these contained the target antigen at a known and stable expression level. Negative controls were set up as tumor tissue slides were evaluated with mouse isotype antibody Ready-to-Use FLEX Negative Control Mouse (Mouse IgG1, IgG2a, IgG2b, IgG3, and IgM, IR750, DAKO, Glostrup, Denmark).

#### 2.2.3. TP53 Mutation Analysis via Sanger Sequencing

*TP53* mutation status was analyzed, as previously described [15]. *TP53* mutations were analyzed using Sanger sequencing. Four exons, 5–8, of the *TP53* gene, were amplified via PCR. The following components of the reaction mixture (total volume 20 μL) were used: 50 ng of DNA; 1 U Taq DNA polymerase (Promega, Fitchburg, WI, USA); 1.5 mM MgCl2; 0.25 mM dATP, dCTP, dGTP, and dTTP; and 0.5 μM primers. Amplification was performed in a GeneAmp PCR System 9700 (Applied Biosystems, Foster City, CA, USA). The primer sequences were as follows: for TP53Ex5_6, forward 5′-CACTTGTGCCCTGACTTTCA-3′ and reverse 5′-CTTAACCCCTCCTCCCAGAG-3′ (product 464 bp); for TP53Ex7, forward 5′-TCATCTTGGGCCTGTGTTATCTC-3′ and reverse 5′-GTGCAGGGTGGCAAGTGG-3′ (163 bp product); and, for TP53Ex8, forward 5′-CAAGGGTGGTTGGGAGTAGA-3′ and reverse 5′-TGCTAGGAAAGAGGCAAGGA-3′ (331 bp product). PCR was performed according to the touchdown protocol for TP53Ex5_6 and TP53Ex8: 94 °C for 5 min, 63 °C for 45 s, 72 °C for 1 min, 1 cycle 94 °C for 1 min, 61 °C for 45 s, 72 °C for 1 min, 34 cycles of 94 °C for 1 min, 60 °C for 45 s, 72 °C for 1 min, and 72 °C for 10 min; the conditions for TP53Ex7 were 94 °C for 5 min, 40 cycles of 94 °C for 30 s, 60 °C for 30 s, 72 °C for 30 s, and 72 °C for 7 min. The PCR products were purified using the MiniElute PCR Purification Kit (Qiagen, Germantown, MD, USA). The sequencing method was carried out using the Big Dye Terminator v1.1 Cycle Sequencing Kit (Applied Biosystems, Foster City, CA, USA) in the GeneAmp PCR System 9700, according to classical sequencing protocols. Nucleotide sequence determination was performed using the same primers used in the initial PCR step. The resulting PCR products were purified to eliminate unincorporated primers and dNTPs using the BigDyeXTerminator Purification Kit. In the next stage, the purified products were separated using the GeneGenetic 3130xl analyzer (Applied Biosystems).

#### 2.2.4. TP53 Promoter Methylation Analysis via Methylation-Specific PCR

The presence of *TP53* promoter methylation was analyzed using MSP. The deamination of the unmethylated cytosine of the *TP53* promoter was performed with sodium bisulfite using the CpGenome DNA kit (Chemicon International Inc., Temecula, CA, USA), according to the manufacturer’s protocol. DNA modified in this way was stored at −80 °C until analyses were carried out. As a positive and negative control, EpiTect methylated/unmethylated human control DNA (bisulfite converted), (Qiagen, Germantown, MD, USA) were used. Methylation-specific PCR for *TP53* promoter methylation was performed as previously described [16,17]. The primers used for methylated CpG island were as follows: 5′-TTCGGTAGGCGGATTATTTG-3′ and 5′-AAATATCCCCGAAACCCAAC-3′. The primers used for unmethylated CpG island were 5′-TTGGTAGGTGGATTATTTGTTT-3′ and 5′-CCAATCCAAAAAAACATATCAC-3′ [17]. In each PCR performed, methylated and unmethylated DNA were included as positive and negative controls. The control for the PCR reaction itself was water. The products obtained from this PCR reaction were separated on 3% agarose gels containing ethidium bromide and documented using the Gel Doc 1000 Bio-Rad Image System. Repeated tests were performed to confirm the results. Methylation-specific PCR for *TP53* promoter methylation was performed on 26 samples due to material unavailability.

#### 2.2.5. P53 Expression Analysis via Immunohistochemistry

P53 protein expression was analyzed via immunohistochemistry. The immunohistochemical expression of p53 was studied using an anti-TP53 monoclonal antibody (clone DO-7, 1:100 dilution; DAKO, Glostrup, Denmark) processed with the EnVision (DAKO) system. The TP53 antibody labels wild-type protein, with a very short half-life and is present in small amounts in normal cells. The mutant-type p53 protein, which is the mutated form, has a longer half-life and is detected via positive staining due to a point mutation. Tumor sections were examined for p53 immunoreactivity under a microscope at 20× and 40× magnifications. The expression of p53 was considered positive when the proportion of positive cells was greater than 10% (Figure 3A). Negative p53 immunostaining resulted in tumor sections (Figure 3B). The IHC results were validated using positive and negative tissue controls in all series of the immunostained slides. Positive controls were used to validate p53 on ovarian carcinoma; these tissues contained the target antigen at a known and stable expression level. Positive endothelial and lymphocyte cells were considered as internal positive controls. The negative controls were set up as neoplastic tissue slides stained using mouse isotype antibody Ready-to-Use FLEX Negative Control Mouse (cocktail of mouse IgG1, IgG2a, IgG2b, IgG3, and IgM, IR750, DAKO, Glostrup, Denmark).

### 2.3. Statistical Analysis

The Shapiro–Wilk test was used to test the distribution of continuous variables. The distribution was found to be abnormal; therefore, these data were presented as medians followed by interquartile ranges (IQRs). Continuous variables were compared using the Mann–Whitney U test. Nominal variables were presented as numbers followed by percentages. Grubbs’ test was used to detect outliers. Spearman’s rank test was used for correlation assessment.

Differences between the categorical variables were evaluated using the Chi2 test or two-tailed Fisher’s exact test. The statistical analysis was exploratory, which is why we did not perform post hoc corrections for multiple testing. For the outcome analyses, overall survival (OS) was defined as the period from diagnosis to the last follow-up, with the censoring of patients who were alive at the last follow-up. Overall survival data were presented as Kaplan–Meier survival curves and compared within subgroups using the log-rank test. Cox hazards regression analyses of OS adjusted for age were performed for each variable.

Additionally, the Multiple Correspondence Analysis (MCA) was employed to evaluate the spatial distribution of the patients according to the TP53 and GATA4 status, expression, and methylation. The analysis was performed using FactoMineR and factoextra R v.4.0.2 packages. *p* values < 0.05 were considered statistically significant.

## 3. Results

### 3.1. Characteristics of Glioblastoma Patients

In this study, we evaluated thirty-one samples of patients with GBM (ten female and twenty-one male). The average age at the time of surgery was 63 ± 7.23 (median = 63; range: 45–76) years old. The median survival time for glioblastoma patients was nine months (range: 0–104). The patients’ characteristics are presented in Table 1.

#### 3.1.1. Methylation and Expression of GATA4 and TP53 Status

The expression of the GATA4 protein was detected in 15 patients (48.4%), while the methylation of *GATA4* promoter was revealed in eight patients (25.8%). GATA4 protein expression was not associated with *GATA4* promoter methylation (Fisher’s exact test *p* = 0.4). A simultaneous lack of GATA4 protein expression and methylation of *GATA4* promoter was found in three patients (9.7%). The percentage intensity of GATA4 expression was similar in *GATA4*-methylated tumors in comparison to *GATA4*-unmethylated tumors (15% vs. 5%, respectively, *p* = 0.58, Figure 4B), although it significantly varied between groups of concomitant statuses of GATA4 protein expression and the methylation of its promoter (Kruskal–Wallis *p* = 0.006; post hoc GATA4 expression (−) and methylation (−) vs. GATA4 expression (+) and methylation (−) *p* = 0.016; GATA4 expression (−) and methylation (−) vs. GATA4 expression (+) and methylation (+) *p* = 0.009; GATA4 expression (−) and methylation (+) vs. GATA4 expression (+) and methylation (−) *p* = 0.04; GATA4 expression (−) and methylation (+) vs. GATA4 expression (+) and methylation (+) *p* = 0.03; Figure 4A). Expression of the p53 protein was detected in eleven (35.5%) patients, *TP53* mutation was detected in six (19.4%) patients, and *TP53* promoter methylation was detected in three (9.7%) patients. However, no association was demonstrated between p53 expression and TP53 mutations (Fisher’s exact test *p* = 0.64), and there was no association between p53 expression and TP53 promoter methylation or between mutation and methylation (Fisher’s exact test, *p* = 1 and *p* = 1, respectively).

#### 3.1.2. Relation between GATA4 Protein Expression and Promoter Methylation and TP53 Mutations, Methylation, and Protein Expression

There was no association between GATA4 protein expression and p53 expression or *TP53* promoter methylation (*p* = 0.27 and *p* = 0.22, respectively). *GATA4* promoter methylation was not associated with p53 expression, TP53 mutation, or promoter methylation (*p* = 0.22, *p* = 0.1 and *p* = 0.51, respectively). There was a strong association between GATA4 expression and *TP53* mutation (*p* = 0.007); patients positive for GATA4 protein expression were positive for *TP53* mutations in six cases in comparison to nine patients without GATA4 protein expression, whereas patients without GATA4 expression were negative for *TP53* mutation in 16 cases, although none of them showed GATA4 expression without a *TP53* mutation.

#### 3.1.3. The Co-Status Results of TP53 and GATA4 Differentiates Prognosis of GBM Patients

The MCA analysis revealed the spatial partitioning of the GBM patients according to various alterations in *TP53* and *GATA4*. The variables’ representation demonstrated two clusters of associations: first—GATA4 expression and mutation status in *TP53* along dimension 1; and second—methylation in *GATA4* promoter and p53 expression along dimension 2, with a total variance of 48.1% (Figure 5A). Disparities in distribution among the individuals could be also observed within the MCA dimensions regarding the aforementioned sets of variables (Figure 5), which then was reflected by the differential OS prognosis (GATA4 expression with *TP53* mutation global *p* = 0.098; Figure 6A; *GATA4* methylation and p53 expression global *p* = 0.0003; Figure 6B). In particular, in the absence of a mutation of *TP53*, the expression of GATA4 was associated with the worst outcome, whereas a lack of both factors resulted in an improved survival rate (HR = 2.51, 95% CI: 1.05–6.01, *p* = 0.038; Table 2). Moreover, the p53 expression without methylation in the *GATA4* promoter correlated with the worst outcomes (HR = 13.1, 95% CI: 2.78–62.2, *p* = 0.001; Table 2).

#### 3.1.4. Survival Analyses Regarding GATA4 and TP53 Status

The median overall survival time in the study group was nine months (range: 0–104). As shown in Figure 6, there was no significant association between GATA4 protein expression and *GATA4* promoter methylation statuses, as well as their cumulative effect and overall survival rate (*p* = 0.19, *p* = 0.17 and *p* = 0.1, respectively, log-rank test). However, in patients positive for the GATA4 protein, p53 expression was associated with the worst outcome (HR = 4.73, 95% CI: 1.61–13.9, *p* = 0.005), but in patients positive for p53 expression, the loss of GATA4 protein expression correlated with improved prognosis (HR = 0.44, 95% CI: 0.13–1.46, *p* = 0.2), and this was related to global significance (*p* = 0.0024, global log-rank test, Figure 6D).

## 4. Discussion

Glioblastoma multiforme (GBM) is the most common primary human brain tumor with high mortality rates, and it exhibits multiple molecular aberrations. Despite multimodal therapy, it has a very unfavorable prognosis, hence the demand for new methods that allow more accurate diagnosis, the prediction of treatment response, and even prevention. In GBM, most *TP53* mutations occur in the DNA-binding domain (DBD), which is encoded by exons 5 to 8, leading to the inhibition of transcription factor activity [12]. Thus, in this study, we analyzed mutations in exons 5–8 of *TP53*. In contrast to a previous study, we found no correlation between *TP53* mutations and p53 expression [16]. The lack of this correlation could be explained by the possibility that an abnormally high level of p53 may be a result of MDM2 dysregulation, even in the absence of *TP53* mutations. MDM2 controls the p53 expression level through a dual mechanism that involves the induction of synthesis and targeting for degradation [18]. The identification of methylated genes may offer a valuable insight into the molecular mechanisms of tumor development and could lead to the identification of new markers of prognostic value. Here, we focused on one of the GATA family transcription factors—*GATA4*. This gene regulates, inter alia, organogenesis, differentiation, proliferation, and apoptosis [19,20,21]. Although it is normally expressed in the brain, where it works as a negative regulator of astrocyte growth, it was found to be absent in 57.7% of GBM samples [4]. In this study, the loss of GATA4 protein expression was observed in 51.6% of the tested patients. *GATA4* silencing can be achieved through promoter hypermethylation or novel somatic mutations associated with the inter alia loss of heterozygosity (LOH) [4]. Our present findings indicate *GATA4* promoter methylation in 25.8% of patients with GBM; this is close to the value of 23.2% found in a previous study on GBM patients [22]. In contrast, it is less common than the sporadic gastric (53.8%) and colorectal cancer (70%) shown in other work [6]. In addition, a similar GATA4 expression level was noted in *GATA4*-methylated tumors and *GATA4*-unmethylated tumors (15% vs. 5%, respectively, *p* = 0.58, Figure 4B), although this level was different between the groups of disparate concomitant GATA4 protein expression and *GATA4* promoter methylation status (Kruskal–Wallis *p* = 0.006, Figure 4A). No significant associations were found between *GATA4* promoter methylation and the expression of the GATA4 protein (Fisher’s exact test *p* = 0.4), indicating that GATA4 protein expression is not regulated by *GATA4* promoter methylation in GBM. The GATA4 level may be regulated by other epigenetic mechanisms, such as microRNA, as well as miR-126, which suppresses GATA4 protein expression [23]. *GATA4* silencing in GBM can also be the result of *GATA4* somatic mutations [4]. *GATA4* mutations have also been demonstrated in breast, ovarian, and lung cancers [24,25,26]. Additionally, *GATA4* promoter methylation has been found to be influenced by patient age, and in one study, the gene methylation ratio varied significantly across different age groups (*p* = 0.027)—77% of patients were older than 60 years [22]. However, no such correlation was found in the present study (*p* = 0.17), possibly due to the small sample size. In addition, GATA4 protein expression did not influence the overall survival rate (*p* = 0.19). Several publications indicate that a high level of GATA4 expression is associated with various outcomes dependent on cancer type; for example, a survival analysis indicated that a high GATA4 expression level is significantly correlated with better OS in second clinical-stage ovarian cancer. However, this assessment was for gene expression, not for protein expression [25]. Another study suggested that a high level of GATA4 expression is correlated with the decreased differentiation of pancreatic cancer (*p* = 0.037). However, these findings from in vitro experiments indicate that GATA4 may function as a negative regulator during carcinogenesis, suggesting that this might result from negative feedback, i.e., the elevation of the expression of GATA4 to inhibit cancerous cell differentiation and proliferation [27]. Our study demonstrates that the concomitant lack of GATA4 protein expression and *GATA4* promoter methylation correlated with better outcomes, although, due to the small size of our study group, these results were insignificant. According to the Gene Transcription Regulation Database (GTRD), *TP53* transcription is regulated by *GATA4*. In childhood ALL, *GATA4* was revealed to regulate p53 through the transcriptional activation of MDM2, thus influencing cell cycle and apoptosis. MDM2 protein expression was increased in cells that ectopically expressed *GATA4*, whereas the decreased expression of p53 was found. The findings indicate that GATA4 potentially modulates apoptosis in ALL via the p53-MDM2 pathway and BCL2. [28]. Surprisingly, p53 expression was associated with the worst outcome in patients positive for GATA4 protein expression (*p* = 0.005); however, in patients positive for p53 expression, the loss of GATA4 protein expression correlated with improved prognosis (*p* = 0.2, although of global significance *p* = 0.0024, Figure 6D). Such dependencies have not been investigated or described in GBM. Additionally, in cases without *TP53* mutations, GATA4 protein expression was associated with the worst outcomes in the first 20 months of observation, whereas a lack of both factors resulted in improved survival (*p* = 0.038). The relationship between *TP53* mutations and *GATA4* expression has been described in one study regarding ovarian cancer; however, it referred to gene expression, not protein expression. However, it has been shown that this dependency was insignificant for *GATA4*. Nevertheless, high levels of expression of other *GATA* family members (*GATA*1-3 and *GATA*6) were linked to a decreased overall survival rate in patients who had *TP53* mutations in ovarian cancer [25]. Finally, the p53 expression without *GATA4* promoter methylation correlated with the worst outcomes. Hence, our data indicate a probable interaction between GATA4 and p53, which may serve as a candidate for the assessment of GBM prognosis. However, this study has several limitations: it had a small sample size, only analyzed mutations of *TP53* within exons 5–8, and did not include an assessment of *GATA4* mutations, which could be used to evaluate additional associations.

## 5. Conclusions

In conclusion, a lack of GATA4 protein expression may not be directly associated with *GATA4* promoter methylation, but the combination of a lack of GATA4 expression and the presence of *GATA4* promoter methylation did not correlate with better outcomes in this study. In addition, GATA4 alterations had no clear prognostic value; however, the loss of GATA4 protein expression was a favorable prognostic marker of survival in patients with increased p53 expression. While the loss of GATA4 protein expression only influenced the outcome in the group with p53 expression, it nevertheless indicated improved prognosis. This indicates that *GATA4* could be an important added prognostic factor in specific subgroups of patients. More research is needed to assess the interaction of GATA4 with TP53 and determine how it affects the development, prognosis, and treatment response in glioblastoma patients.

## Figures and Tables

**Figure 1 genes-14-01146-f001:**
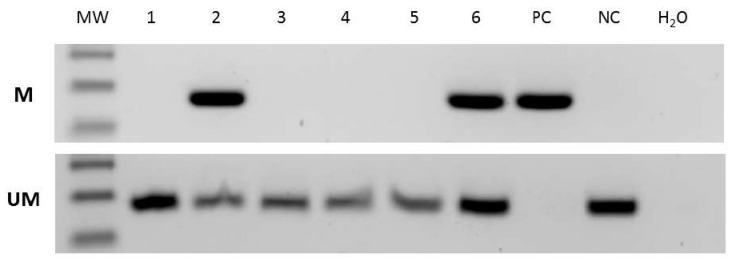
Representative *GATA4* promoter methylation results via MS-PCR. The product size of the methylated and unmethylated amplicons was 136 bp and 142 bp, respectively. Samples 1, 3, 4, and 5 are unmethylated; samples 2 and 6 are methylated. Abbreviations: M, methylated PCR product; U, unmethylated PCR product; PC, positive control—methylated control; NC, negative control—unmethylated control; MW, molecular-weight marker.

**Figure 2 genes-14-01146-f002:**
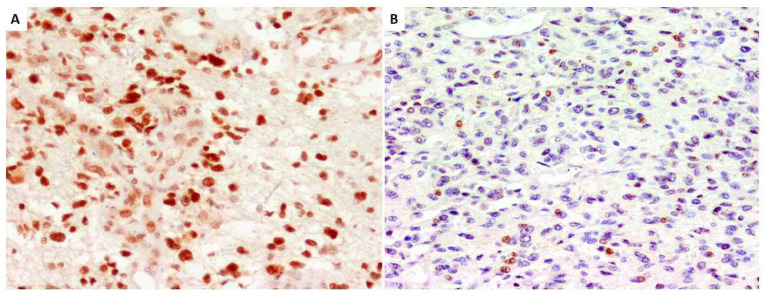
Strong GATA-4 protein nuclear expression in most glioblastoma cells, magnification of 200× (**A**); example of sparse expression of GATA-4 protein (classified as negative) in glioblastoma, magnification 200× (**B**) (anti-GATA-4 antibody, Santa Cruz Biotechnology).

**Figure 3 genes-14-01146-f003:**
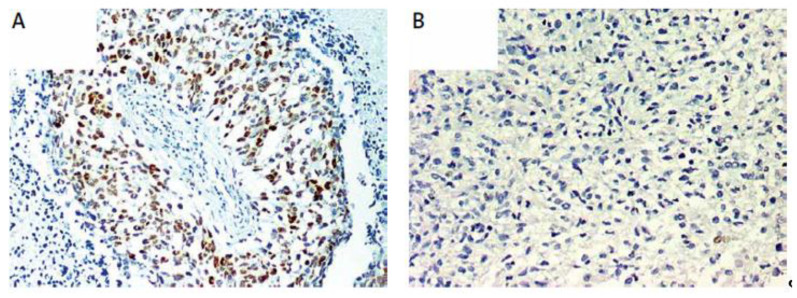
(**A**) Immunohistochemically detected p53 overexpression. (**B**) Sample negative for p53 overexpression.

**Figure 4 genes-14-01146-f004:**
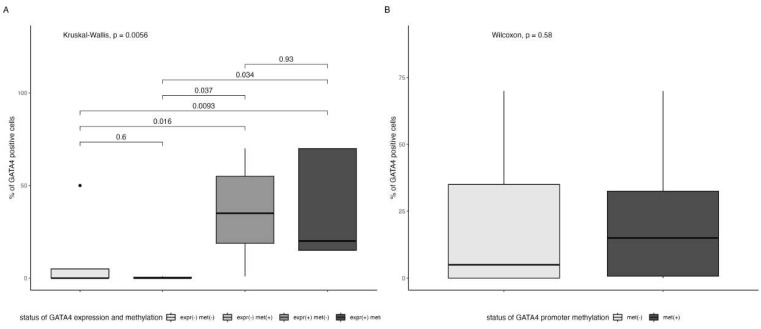
Comparison of GATA4-positive cells regarding *GATA4* promoter methylation in groups of concomitant statuses of GATA4 protein expression and methylation of its promoter (**A**). Comparison of GATA4-positive cells regarding *GATA4* promoter methylation (**B**).

**Figure 5 genes-14-01146-f005:**
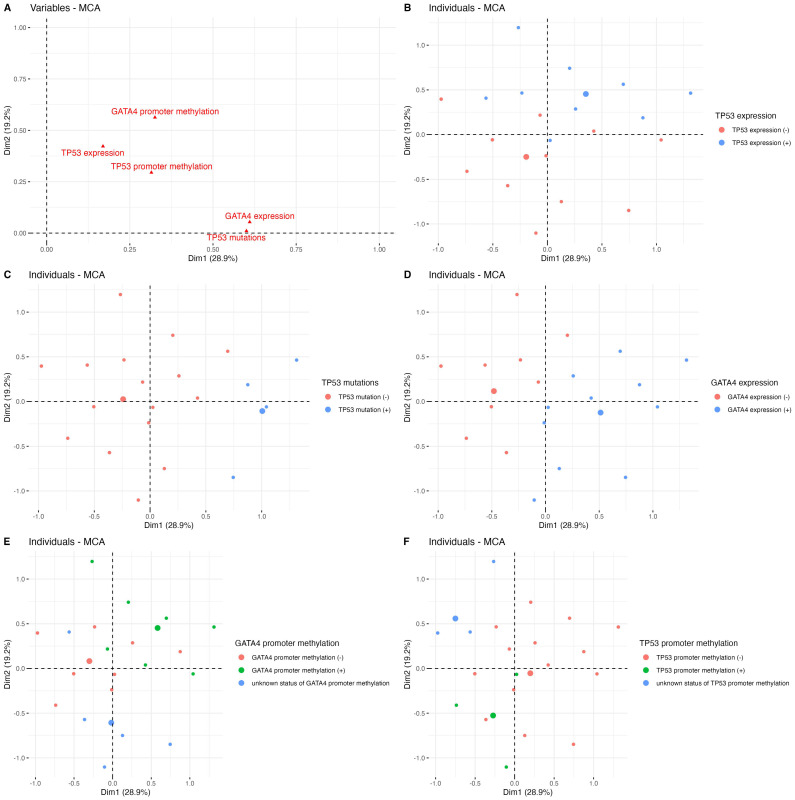
The Multiple Correspondence Analysis (MCA) revealed various associations between GATA4 and TP53. In particular, the variables’ representation showed high concordance in the resultant effects of the GATA4 expression and mutation status in *TP53* along dimension 1 and between methylation in *GATA4* promoter and p53 expression along dimension 2 (**A**), with a total variance of 48.1%. As demonstrated, the spatial partitioning of individuals originated from the total effects exerted by all of these variables considered together, which clearly clustered them according to p53 expression (**B**), mutations in *TP53* (**C**), GATA4 expression (**D**), and methylation of *GATA4* (**E**) and *TP53* promoters (**F**), respectively.

**Figure 6 genes-14-01146-f006:**
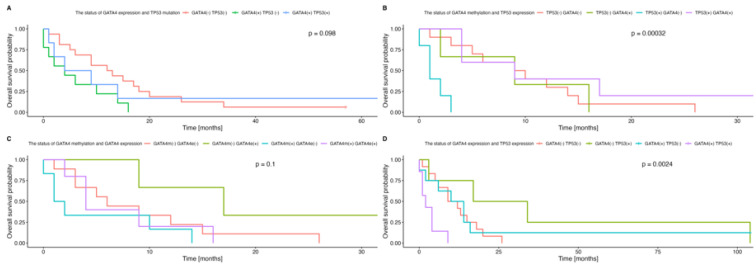
Kaplan–Meier curves demonstrating differential prognoses of the overall survival (OS) of the GBM patients with regard to the cumulative effects of GATA4 expression and *TP53* mutation (**A**), *GATA4* promoter methylation and p53 expression (**B**), *GATA4* promoter methylation and GATA4 expression (**C**), and GATA4 and p53 expression (**D**).

**Table 1 genes-14-01146-t001:** Study group characteristics. Nominal variables are presented as numbers followed by percentages, and continuous variables are presented as medians followed by interquartile ranges.

Variable	Number of Positive Cases (%) or Median (IQR)	Number of Patients with Available Data
Sex (males)	21 (67.74%)	31
GATA4 expression (IHC)	15 (48.39%)	31
*GATA4* methylation (PCR)	8 (25.8%)	23
p53 expression (IHC)	11 (35.5%)	31
*TP53* mutations (IHC)	6 (19.4%)	31
*TP53* methylation	3 (9.7%)	26
Mean age at diagnosis (years)	63 ± 7.23 (median = 63; range 45–76)	31
Overall survival (months)	9 (range 0–104)	31

**Table 2 genes-14-01146-t002:** The summary of Cox proportional hazards models according to clinical variables.

Clinical Variable	HR (95% CI)	*p*-Value
GATA4 expression and *TP53* mutation		
GATA4 (−) *TP53* (−)	ref	
GATA4 (+) *TP53* (−)	2.51 (1.05–6.01)	0.038
GATA4 (+) *TP53* (+)	1.24 (0.45–3.42)	0.7
p53 expression and *GATA4* methylation		
p53 (−) *GATA4* (−)	ref	
p53 (−) *GATA4* (+)	1.1 (0.3–4.08)	0.9
p53 (+) *GATA4* (−)	13.1 (2.78–62.2)	0.001
p53 (+) *GATA4* (+)	0.6 (0.18–1.96)	0.4
*GATA4* methylation and GATA4 expression		
*GATA4*m (−) GATA4 (−)	ref	
*GATA4*m (−) GATA4 (+)	0.29 (0.06–1.41)	0.13
*GATA4*m (+) GATA4 (−)	2.08 (0.7–6.16)	0.2
*GATA4*m (+) GATA4 (+)	1.27 (0.41–3.91)	0.7
GATA4 expression and p53 expression		
GATA4 (−) p53 (−)	ref	
GATA4 (−) p53 (+)	0.44 (0.13–1.46)	0.2
GATA4 (+) p53 (−)	0.77 (0.29–2.03)	0.6
GATA4 (+) p53 (+)	4.73 (1.61–13.9)	0.005

## Data Availability

The data presented in this study are available upon request from the corresponding author.

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
