# Peer review of "Potential Prognostic Value of GATA4 Depends on the p53 Expression in Primary Glioblastoma Patients"

_genes, 2023, doi:10.3390/genes14061146_

Round 1

Reviewer 1 Report

The current work investigates the relationship among GATA4 protein expression, GATA4's promoter methylation, TP53 expression, TP53 mutations and TP53's promoter methylation in glioblastoma. All the data were derived from experiments applied in clinical samples coming from glioblastoma patients using classical methods such as immunohistochemistry, Sanger sequencing and methylation-specific PCR. 

Very little information can be found in the existing literature concerning the role of GATA4 in glioblastoma and how it is correlated with TP53, a crucial gene in this cancer type. After performing various statistical analyses, the authors conclude that the correlation beteen GATA4 and TP53 expression has potentially a prognostic value. Specifically, TP53 expression accompanied by GATA4 expression lead to a worst outcome while when GATA4 expression is missing the patients survival is better. Another significant finding is that GATA4 expression is not correlated with the methylation status of its promoter showing that the regulation of its expression is complexed and mediated by additional factors. 

Taking into consideration the above findings, the initial hypothesis and the methodological approach, the manuscript is consistent with the aims and the scope of the journal. Overall, the findings are novel and they willl be useful for the journal's readers.

However, I believe that the following corrections would improve the manuscript:

1. The Materials and Methods part needs to be restructured so that it will be easier for the reader to follow it. For example: 2.3.3 can be placed after 2.3.1 and 2.3.2 can be divided in three separate sections, so that each technique for TP53 will be placed separetely accompanied by a specific title e.g Methylation-specific PCR for TP53 promoter etc.

2. In 2.3.2 Material and Methods section, the authors should provide the primer sequences for the methylated and unmethylated TP53 promoters.

3. Could the authors also provide represantative immunohistochemistry images for TP53 (wild and mutated) as they did for GATA4?

4. Could the authors refer to the software they used for the analysis of the immunohistochemistry images? How did they perform the staining quantitation?

5. In lines 67-69 the reference/s is/are missing.

6. In lines 354-355 citation is missing.

7. Τext size and resolution in Figures 3,4,5 should be increased. Particularly in Figure 5 the text is so small that makes it impossible to read the legends and the axes.

8. According to Table 1, the data available for the methylation status of the GATA4 and TP53 promoters come from 23 and 26 patients, respectively. Why not from all the 31 patients? The authors should clarify this in the methods or main part of the manuscript.

9. In section 3.2.4, the authors should provide the HR for the improved prognosis.

Overall, the english language is appropriate and understandable. 

It is strongly suggested that the authors should proofread the manuscript for errors. 

For example:

1. Lines 77-79: The sentence needs to be corrected e.g In GBM, TP53 mutations are mostly point mutations which not only…

2. Lines: 79-80: It should also be noted...

Author Response

Dear Reviewer,

We highly appreciate your kindly reviews and comments how to revise our manuscript titled ”Potential prognostic value of GATA4 depends on the p53 expression in primary glioblastoma patients.” and thank you for the opportunity to improve them. We followed all your suggestions and revised the manuscript accordingly. Our specific comments to your concerns are included below.

We appreciate the time and effort that you have dedicated to providing your valuable feedback on our manuscript. We hope that the provided amendments improved the manuscript accordingly to meet Your requirements.

With this letter, we are submitting the revised manuscript and Figure 3.

  1. The Materials and Methods part needs to be restructured so that it will be easier for the reader to follow it. For example: 2.3.3 can be placed after 2.3.1 and 2.3.2 can be divided in three separate sections, so that each technique for TP53 will be placed separately accompanied by a specific title e.g Methylation-specific PCR for TP53 promoter etc.

According to the Reviewer's suggestion, the Materials and Methods part has been restructured - 2.3.3 has been moved after 2.3.1, and 2.3.2 has been divided into three separate sections so that each technique for TP53 is placed separately with specific titles.

  1. In 2.3.2 Material and Methods section, the authors should provide the primer sequences for the methylated and unmethylated TP53 promoters.

We did not include primer sequences for the methylated and unmethylated TP53 promoters as we referred to the previous studies where these primers were applied. However, according to your suggestion, we added the primers used for methylated CpG island were 5′-TTCGGTAGGCGGATTATTTG-3′ and 5′-AAATATCCCCGAAACCCAAC-3′. Primers for unmethylated CpG island were 5′-TTGGTAGGTGGATTATTTGTTT-3′ and 5′-CCAATCCAAAAAAACATATCAC-3′ and incorporated them into the manuscript [1].

[1] Amatya VJ, Naumann U, Weller M, Ohgaki H. TP53 promoter methylation in human gliomas. Acta Neuropathol 2005;110:178–84. https://doi.org/10.1007/s00401-005-1041-5.

  1. Could the authors also provide represantative immunohistochemistry images for TP53 (wild and mutated) as they did for GATA4?

Yes, a representative image was published in Jesionek-Kupnicka D, Braun M, Trąbska-Kluch B, Czech J, Szybka M, Szymańska B, Kulczycka-Wojdala D, Bieńkowski M, Kordek R, Zawlik I. MiR-21, miR-34a, miR-125b, miR-181d and miR-648 levels inversely correlate with MGMT and TP53 expression in primary glioblastoma patients. Arch Med Sci. 2019 Mar;15(2):504-512. doi: 10.5114/aoms.2017.69374. Epub 2017 Jul 31. PMID: 30899304; PMCID: PMC6425218:

Figure 3. A – Immunohistochemically detected p53 protein overexpression (mutated TP53).
B – Sample negative for p53 overexpression (wild-type TP53).

  1. Could the authors refer to the software they used for the analysis of the immunohistochemistry images? How did they perform the staining quantitation?

As routinely, no software was used for analysis of the immunohistochemistry images. Presence or absence of the GATA4 and p53 expression was assessed by two independent experienced surgical pathologists (MB, DJK). The expression was considered positive when >10% of cells revealed nuclear positivity.

  1. In lines 67-69 the reference/s is/are missing.

Thank you for the remark of missing reference however citation number [5] refers to the entire text contained in lines 67-72.

  1. In lines 354-355 citation is missing.

In lines 354-355 the data is taken from the Gene Transcription Regulation Database (GTRD). Therefore it is not possible to use citation here.

  1. Τext size and resolution in Figures 3,4,5 should be increased. Particularly in Figure 5 the text is so small that makes it impossible to read the legends and the axes.

Thank you for your feedback in regards to the size and quality of the figures. According to Reviewer’s guidance we increased the size of all images as much as possible. Figures 5 (previously Fig. 4) is a collective panel and it is not possible to separate it into individual figures.

  1. According to Table 1, the data available for the methylation status of the GATA4 and TP53 promoters come from 23 and 26 patients, respectively. Why not from all the 31 patients? The authors should clarify this in the methods or main part of the manuscript.

We do apologize that we did not include the information why the samples size was reduced according to Table 1 in the Manuscript. In agreement with the Reviewer's suggestion we corrected the sentence as follows: “Methylation-specific PCR for TP53 promoter methylation was performed on 26 samples due to unavailability of material.” and “Methylation-specific PCR for GATA4 was performed on 23 samples due to unavailability of material.” in the Materials and Methods.

  1. In section 3.2.4, the authors should provide the HR for the improved prognosis.

According to the Reviewer's comment, we added the HR value and corrected the sentence as follows: “However, in patients positive for GATA4 protein, p53 expression was associated with the worst outcome (HR=4.73, 95% CI: 1.61-13.9, p=0.005) , but in patients positive for p53 expression, loss of GATA4 protein expression correlated with improved prognosis (HR=0.44, 95% CI: 0.13-1.46, p=0.2), although of global significance (p=0.0024, global log-rank test, Figure 6D).”

Reviewer 2 Report

This study investigates the relationship between tumour suppressor p53 and  a putative tumor suppressor transcription factor GATA4, in glioblastoma (GB). Although the diagnostic value of p53 has long been known and used in GB diagnostics the potential association between p53 and GATA4 is an interesing question with potential clinical relevance. The authors have  correlated TP53 mutation status, GATA4 expression and promoter methylation status with survival data, in a cohort of 31 patients with primary GB. The main finding is that a potential prognostic power of GATA4 requres consideration of the p53 status. This finding is both interesting and potentially clinically relevant. However, the manuscript as it stands now  has a number of insufficiancies that make it unsuitable for publishing in the current form. 

Major concerns:

1) The conclusion about p53 status is based on sequencing data restricted to exons 5-8 of the TP53 gene. Eventhough this region accommodates p53 hotspot mutations it cannot be excluded that some less common mutations  residing outside of sequenced regions have been been missed.

2) p53 expression was also analyzed by immunohistochemical (IHC) staining using antibody DO7. The conclusion about p53 status was made on the basis of  signal itensity and erroneous presumption that DO7 "labels wild-type p53" (line 159). As DO7 antibody does not discriminate between wild type and mutant p53 forms conclusions about p53 mutational status based on DO7 patterns may be misleading. These shortcomings may explain lack of correlation between p53 expression and TP53 mutations. 

3) The conclusion about lack of association between p53 expression and TP53 mutations (within exons 5-8) does not take into consideration the possibility that abnormally high steady state levels of p53 may also be a result of MDM2 overexpression even in the absence of p53 mutations. This possibility needs to be addressed.   

4) Lack of relationship between GATA4 expression assessed by IHC and methylation status of the GATA4 promoter is explained by "other epigenetic mechanisms" (lines 338-339). In the absense of any evidence that would  support the impact of some "other epigenetic mechanisms" such explanation is less than satisfatory. Especially considering that there is  a  well defined mechanism for GATA4 silencing by mutations in the GATA4 gene. Lack of information about GATA4 mutational status is a serious  drawback that needs to be addressed. 

5) The manuscript is poorly prepared and written. Figs. 4 and 5 are simply unreadable precluding any possibility to evaluate the data they show. Important statements are unsupported by relevant references. For example,  statement that "loss of GATA4 is associated with a shorter survival rate in GB patients" (lines 68-69). Usage of non-standard terms and ackward formulations is another insufficiency. 

To name just a few: "WHO G IV" (lane 48) >>> should be  WHO°IV ; "TP53 protein", "TP53 antibody" >>> TP53 refers to the gene not protein; "Routine treatment covers surgical resection followed by radiochemotherapy (RT-CHT) and adjuvant chemotherapy with temozolomide" (lanes 53-54) >>>  TMZ is the only standardly accepted chemotherapy for GB; "... median survival of 14.6-15 months in highly selected patients". 

poor

Author Response

Dear Reviewer,

We highly appreciate your kindly reviews and comments how to revise our manuscript titled ”Potential prognostic value of GATA4 depends on the p53 expression in primary glioblastoma patients.” and thank you for the opportunity to improve them. We followed all your suggestions and revised the manuscript accordingly. Our specific comments to your concerns are included below.

We appreciate the time and effort that you have dedicated to providing your valuable feedback on our manuscript. We hope that the provided amendments improved the manuscript accordingly to meet Your requirements.

With this letter, we are submitting the revised Manuscript.

1) The conclusion about p53 status is based on sequencing data restricted to exons 5-8 of the TP53 gene. Even though this region accommodates p53 hotspot mutations it cannot be excluded that some less common mutations  residing outside of sequenced regions have been missed.

We agree with your observation that some less common mutations residing outside of sequenced regions have been not considered in our study. In GBM most TP53 mutations occur within the DNA binding domain (DBD) which is encoded by exons 5 to 8 leading to inhibition of transcription factor activity [1]. Thus in this study we analyzed mutations in exons 5-8 of TP53. We added this explanation in the Discussion for elucidation.

[1] Zhang Y, Dube C, Gibert M, Cruickshanks N, Wang B, Coughlan M, et al. The p53 Pathway in Glioblastoma. Cancers 2018;10:297. https://doi.org/10.3390/cancers10090297.

2) p53 expression was also analyzed by immunohistochemical (IHC) staining using antibody DO7. The conclusion about p53 status was made on the basis of  signal intensity and erroneous presumption that DO7 "labels wild-type p53" (line 159). As DO7 antibody does not discriminate between wild type and mutant p53 forms conclusions about p53 mutational status based on DO7 patterns may be misleading. These shortcomings may explain lack of correlation between p53 expression and TP53 mutations. 

P53 staining using the selected clone is routinely used in diagnostic pathology to differentiate between reactive changes and neoplasms, but also as a valid method for distinguishing between wild-type and mutated TP53 status [1-3]. Normally, wild-type p53 protein has low expression levels in normal cells, and it is often undetectable by immunohistochemistry due to its very short half-life and small amounts in cells [4-6]. Conversely, mutant-type p53 protein significantly prolongs the half-life of the protein, making it detectable by positive staining in immunohistochemistry, as demonstrated by the chosen product in this study [4-6].

  1. Nguyen JK, Przybycin CG, McKenney JK, Magi-Galluzzi C. Immunohistochemical staining patterns of Ki-67 and p53 in florid reactive urothelial atypia and urothelial carcinoma in situ demonstrate significant overlap. Hum Pathol. 2020 Apr;98:81-88. doi: 10.1016/j.humpath.2020.02.008. Epub 2020 Mar 3. PMID: 32142835.
  2. Hutchings D, Waters KM, Weiss MJ, Wolfgang CL, Makary MA, He J, Cameron JL, Wood LD, Hruban RH. Cancerization of the Pancreatic Ducts: Demonstration of a Common and Under-recognized Process Using Immunolabeling of Paired Duct Lesions and Invasive Pancreatic Ductal Adenocarcinoma for p53 and Smad4 Expression. Am J Surg Pathol. 2018 Nov;42(11):1556-1561. doi: 10.1097/PAS.0000000000001148. PMID: 30212393; PMCID: PMC6266304.
  3. Sung YN, Kim D, Kim J. p53 immunostaining pattern is a useful surrogate marker for TP53 gene mutations. Diagn Pathol. 2022 Dec 5;17(1):92. doi: 10.1186/s13000-022-01273-w. PMID: 36471402; PMCID: PMC9720942
  4. Haupt Y, Maya R, Kazaz A, Oren M. Mdm2 promotes the rapid degradation of p53. Nature. 1997 May 15;387(6630):296-9. doi: 10.1038/387296a0. PMID: 9153395.
  5. Maki CG, Huibregtse JM, Howley PM. In vivo ubiquitination and proteasome-mediated degradation of p53(1). Cancer Res. 1996 Jun 1;56(11):2649-54. PMID: 8653711.
  6. Giaccia AJ, Kastan MB. The complexity of p53 modulation: emerging patterns from

divergent signals. Genes Dev. 1998 Oct 1;12(19):2973-83. doi:10.1101/gad.12.19.2973. PMID: 9765199.

3) The conclusion about lack of association between p53 expression and TP53 mutations (within exons 5-8) does not take into consideration the possibility that abnormally high steady state levels of p53 may also be a result of MDM2 overexpression even in the absence of p53 mutations. This possibility needs to be addressed.   

We admit that we have not considered the possible relationship between abnormally high steady state levels of p53 and MDM2 overexpression despite the absence of TP53 mutations. We have omitted the description of these correlation because they were not evaluated in this paper. However, following the Reviewer's advice, we added this data to the Discussion part as follow: “The lack of this correlation could be explain by the possibility that abnormally high level of p53 may be a result of MDM2 dysregulation even in the absence of TP53 mutations. MDM2 controls the p53 expression level through a dual mechanism that involves induction of synthesis and targeting for degradation [19].”

4) Lack of relationship between GATA4 expression assessed by IHC and methylation status of the GATA4 promoter is explained by "other epigenetic mechanisms" (lines 338-339). In the absence of any evidence that would  support the impact of some "other epigenetic mechanisms" such explanation is less than satisfactory. Especially considering that there is  a  well-defined mechanism for GATA4 silencing by mutations in the GATA4 gene. Lack of information about GATA4 mutational status is a serious  drawback that needs to be addressed. 

Yes, we agree. In Discussion we added that “GATA4 level may be regulated by other epigenetic mechanisms, such as microRNA for example miR-126 that suppressed GATA4 protein expression [26]. GATA4 silencing in GBM can be also result of GATA4 somatic mutations [5].“ We also agree that detailed research is needed in this field but from this point of view we are unable to assess it in our study. In the Discussion we also added the limitations of this study.

5) The manuscript is poorly prepared and written. Figs. 4 and 5 are simply unreadable precluding any possibility to evaluate the data they show. Important statements are unsupported by relevant references. For example,  statement that "loss of GATA4 is associated with a shorter survival rate in GB patients" (lines 68-69). Usage of non-standard terms and ackward formulations is another insufficiency. 

Thank you for Your feedback in regards to the size and quality of the figures. According to Reviewer’s guidance we increased the size of all images as much as possible. Figures 5 (previously Fig. 4) is a collective panel and it is not possible to separate it into individual figures. Thank you for the remark to missing reference, however citation number [5] refers to the entire text contained in lines 67-72. The text has been corrected to make it more readable.

To name just a few: "WHO G IV" (lane 48) >>> should be  WHO°IV ; "TP53 protein", "TP53 antibody" >>> TP53 refers to the gene not protein; "Routine treatment covers surgical resection followed by radiochemotherapy (RT-CHT) and adjuvant chemotherapy with temozolomide" (lanes 53-54) >>>  TMZ is the only standardly accepted chemotherapy for GB; "... median survival of 14.6-15 months in highly selected patients". 

All errors and shortcomings were corrected and underwent repeated and intensive language correction.  Moreover, the name of the substance temozolomide in the sentence "Routine treatment covers surgical resection followed by radiochemotherapy (RT-CHT) and adjuvant chemotherapy with temozolomide" (lanes 53-54) was used intentionally to highlight its role in the multimodality treatment of GBM.

Round 2

Reviewer 2 Report

while most of the issues raised during the 1st round have been addressed more or less satisfactorily there afe still some essential deficiencies that preclude this manuscript acceptance 

Inappropriate citations.

1) to support their statements that "...loss of GATA4 protein expression is linked to a decreased survival rate oin GBM patients..." (line 78) and GATA4  "...was found to be absent in 57.6% to 70% of GBM samples" (line 443) the authors cite the study by Agnihotri et al, in which the association between GATA4 and GBM survival was not addressed at all. This  is not only misleading it is incorrect. 

2) the authors cite the study by VaitkienÄ— P et al, to support their statement that GATA4  "...was found to be absent in 57.6% to 70% of GBM samples" (line 443). However, the cited study by VaitkienÄ— P et al has not analyzed GATA4 expression at all but only the methylation status of GATA4 promoter with the major finding being that GATA4 promoter is abnormally methylated in about 23% of GBM samples analyzed.

3) very same concern applies to references 7 and 11 cited to support the statement that GATA4  "...was found to be absent in 57.6% to 70% of GBM samples" (line 443). Ref 7 is a study concerned with colorectal and gastric cancer and zero relation to GBM whereas ref 11 concerns major pathways in GBM not individual putative markers like GATA4.

Miscitations as above are incompatible with basic research ethics and preclude this manuscript acceptance for publishing 

acceptable

Author Response

Brief Summary:

While most of the issues raised during the 1st round have been addressed more or less satisfactorily there are still some essential deficiencies that preclude this manuscript acceptance.

Dear Reviewer,

We sincerely apologize for the mistakes in our manuscript titled ”Potential prognostic value of GATA4 depends on the p53 expression in primary glioblastoma patients.” and thank you for the opportunity to correct them.

We appreciate the time and effort that you have dedicated to providing your valuable feedback on our manuscript. We hope that the provided amendments improved the manuscript accordingly to meet your requirements.

Corrections:

Inappropriate citations.

  • to support their statements that "...loss of GATA4 protein expression is linked to a decreased survival rate oin GBM patients..." (line 78) and GATA4 "...was found to be absent in 57.6% to 70% of GBM samples" (line 443) the authors cite the study by Agnihotri et al, in which the association between GATA4 and GBM survival was not addressed at all. This  is not only misleading it is incorrect.

Thank you for pointing out the mistakes that appeared in the lines 78 and 443. According to the Reviewer suggestion we have corrected sentence in line 78 as follows: “Agnihotri et al. demonstrated that the loss of GATA4 protein expression was a negative survival prognostic factor in GBM patients and that GATA4 can sensitize GBM cells to alkylating drugs [4]”. In line 443 we have corrected the sentence as follows: „Although it is normally expressed in the brain, where it works as a negative regulator of astrocyte growth, it was found to be absent in 57.7% of GBM samples [4].”

4. Agnihotri S, Wolf A, Munoz DM, Smith CJ, Gajadhar A, Restrepo A, et al. A GATA4-regulated tumor suppressor network represses formation of malignant human astrocytomas. Journal of Experimental Medicine 2011;208:689–702. https://doi.org/10.1084/jem.20102099.

  • the authors cite the study by VaitkienÄ— P et al, to support their statement that GATA4 "...was found to be absent in 57.6% to 70% of GBM samples" (line 443). However, the cited study by VaitkienÄ— P et al has not analyzed GATA4 expression at all but only the methylation status of GATA4 promoter with the major finding being that GATA4 promoter is abnormally methylated in about 23% of GBM samples analyzed.

We agree with Reviewer that in the study that we cited, VaitkienÄ— P et al has not analyzed GATA4 expression at all but only the methylation status of GATA4 promoter with the major finding being that GATA4 promoter is abnormally methylated in about 23% of GBM samples. We have corrected this omission and removed this citation from the line number 443.

  • very same concern applies to references 7 and 11 cited to support the statement that GATA4 "...was found to be absent in 57.6% to 70% of GBM samples" (line 443). Ref 7 is a study concerned with colorectal and gastric cancer and zero relation to GBM whereas ref 11 concerns major pathways in GBM not individual putative markers like GATA4.

We are so sorry for this shortcoming with the incorrect numbering of citations. According to the Reviewer suggestion we have corrected these errors and removed the incorrect citations from line 443. We originally focused on supporting only association, but we agree that it should be presented in a broader context, because these were studies other than GBM.
